# Recurrent gain of function mutation in calcium channel *CACNA1H* causes early-onset hypertension with primary aldosteronism

Ute I Scholl[1,2], Gabriel Stölting[3], Carol Nelson-Williams[1], Alfred A Vichot[1], Murim Choi[1,4], Erin Loring[1,4], Manju L Prasad[5], Gerald Goh[1], Tobias Carling[6], C Christofer Juhlin[6,7], Ivo Quack[2], Lars C Rump[2], Anne Thiel[2], Marc Lande[8], Britney G Frazier[9], Majid Rasoulpour[10], David L Bowlin[11], Christine B Sethna[12], Howard Trachtman[13], Christoph Fahlke[3], Richard P Lifton[1,4]*

[1]Department of Genetics, Howard Hughes Medical Institute, Yale University School of Medicine, New Haven, United States; [2]Division of Nephrology, Heinrich Heine University Düsseldorf, Düsseldorf, Germany; [3]Institute of Complex Systems, Zelluläre Biophysik, Forschungszentrum Jülich, Jülich, Germany; [4]Yale Center for Mendelian Genomics, New Haven, United States; [5]Department of Pathology, Yale University School of Medicine, New Haven, United States; [6]Yale Endocrine Neoplasia Laboratory, Yale School of Medicine, New Haven, United States; [7]Department of Oncology-Pathology, Karolinska Institutet, Karolinska University Hospital, Stockholm, Sweden; [8]Division of Pediatric Nephrology, University of Rochester Medical Center, Rochester, United States; [9]Madigan Army Medical Center, Tacoma, United States; [10]Connecticut Children's Medical Center, Hartford, United States; [11]Intermed Consultants Ltd, Edina, United States; [12]Department of Pediatrics, Cohen Children's Medical Center of New York, New Hyde Park, United States; [13]Department of Pediatrics, NYU Langone Medical Center, New York, United States

*For correspondence: richard. lifton@yale.edu

**Abstract** Many Mendelian traits are likely unrecognized owing to absence of traditional segregation patterns in families due to causation by de novo mutations, incomplete penetrance, and/or variable expressivity. Genome-level sequencing can overcome these complications. Extreme childhood phenotypes are promising candidates for new Mendelian traits. One example is early onset hypertension, a rare form of a global cause of morbidity and mortality. We performed exome sequencing of 40 unrelated subjects with hypertension due to primary aldosteronism by age 10. Five subjects (12.5%) shared the identical, previously unidentified, heterozygous *CACNA1H*^M1549V mutation. Two mutations were demonstrated to be de novo events, and all mutations occurred independently. *CACNA1H* encodes a voltage-gated calcium channel (Ca$_V$3.2) expressed in adrenal glomerulosa. CACNA1H^M1549V showed drastically impaired channel inactivation and activation at more hyperpolarized potentials, producing increased intracellular Ca$^{2+}$, the signal for aldosterone production. This mutation explains disease pathogenesis and provides new insight into mechanisms mediating aldosterone production and hypertension.

## Introduction

The steroid hormone aldosterone is normally produced in the adrenal zona glomerulosa in response to either angiotensin II, which is produced in response to volume depletion, or hyperkalemia (high plasma K$^+$ level). Both stimuli cause membrane depolarization, activating voltage-gated Ca$^{2+}$ channels;

**eLife digest** The consequence of mutations to the large majority of human genes is unknown. Most mutations that are currently known were discovered by tracing their effects through families. This allows the locations of mutations to be pinpointed on chromosomes—the structures that genetic material is packaged into. Other mutations are harder to trace because individuals with these mutations may develop very different signs and symptoms, or not develop clinical abnormalities at all. Alternatively, a trait may appear sporadically in a family because the mutation arises anew in the affected subject.

Recently developed technologies that allow scientists to rapidly sequence all the gene-encoding regions of an individual's DNA—their genome—offer a new way to identify harmful genetic variants. Comparing the genomes of individuals with rare disorders can reveal if the individuals share any genetic mutations in common that could cause their symptoms.

Scholl et al. used this strategy to sequence the genomes of 40 individuals with a rare type of hypertension—a condition that causes high blood pressure, and increases the risk of strokes, kidney failure and heart attacks—that develops early in childhood. In this form of the disease, high blood pressure is caused by the adrenal glands above the kidneys producing too much of a hormone called aldosterone. Some genetic causes of this form of the disease have already been identified. Now, Scholl et al. have found a new genetic mutation present in five families with this condition. Two of the individuals were the first in their families to develop this mutation, while three others inherited it. Some of the family members with this mutation had hypertension and some did not.

The mutation is in a gene that encodes a type of calcium channel—a protein found in the membrane that surrounds cells, and which can open and close to control the amount of calcium in the cell. This particular calcium channel is abundant in the cells of the adrenal gland. Scholl et al. found that the mutation causes the calcium channels to be more likely to open and take longer to close. This increases the number of calcium ions that move into the cell, which causes the adrenal gland to produce more aldosterone. These new insights have provided a new way of diagnosing early-onset hypertension, and suggest that targeting calcium channels could help to develop new treatments for this disease.

increased intracellular $Ca^{2+}$ provides the signal that triggers aldosterone production (*Spät and Hunyady, 2004*). In the setting of volume depletion, aldosterone signaling in renal and intestinal epithelia produces increased salt (re)absorption, promoting restoration of intravascular volume; in hyperkalemia, aldosterone promotes increased potassium secretion, restoring electrolyte balance.

Pathological secretion of aldosterone in the absence of normal physiological stimuli leads to primary aldosteronism (PA), producing increased salt (re)absorption and hypertension. Hypokalemia is a frequently associated finding, resulting from increased renal $K^+$ elimination. PA is found in 10% of patients referred for evaluation of hypertension (*Conn, 1955*; *Rossi et al., 2006*). About half of these patients have adrenal aldosterone-producing adenomas (APAs). Germline mutations in three genes have been shown to cause rare Mendelian forms of early-onset PA. Gene fusions leading to constitutive expression of aldosterone synthase (encoded by *CYP11B2*), a rate-limiting enzyme in aldosterone biosynthesis, cause Glucocorticoid-Remediable Aldosteronism (GRA) (*Lifton et al., 1992*). Mutations in and near the selectively filter of the $K^+$ channel encoded by *KCNJ5* result in channels that conduct $Na^+$, leading to adrenal glomerulosa cell depolarization and activation of $Ca^{2+}$ channels, producing a Mendelian form of aldosteronism (*Choi et al., 2011*). Gain of function mutations in the calcium channel encoded by *CACNA1D* cause increased $Ca^{2+}$ channel activity and another form of PA. These latter patients also have seizures, neurodevelopmental and neuromuscular abnormalities owing to gain of function effects of *CACNA1D* in the nervous system (*Scholl et al., 2013*). Families with GRA often have many affected subjects and were identified by linkage analysis in extended families (*Lifton et al., 1992*). Germline mutations in *KCNJ5* are typically de novo or in small nuclear families; similarly, *CACNA1D* mutations to date are all de novo (*Choi et al., 2011*; *Scholl et al., 2012, 2013*). Germline mutations in *KCNJ5* and *CACNA1D* were found following identification of the same or related somatic mutations as drivers of APAs (*Choi et al., 2011*; *Scholl et al., 2012*; *Azizan et al., 2013*; *Scholl et al., 2013*).

The causes of PA in many patients remain undetermined. Although Mendelian inheritance has been suggested by recurrence of PA in some kindreds without mutations in known genes (*Stowasser et al., 1992*; *Torpy et al., 1998*; *Lafferty et al., 2000*), traditional linkage analysis has failed to identify additional causative genes, likely due to a combination of factors including locus heterogeneity, high frequency of de novo mutations, reduced penetrance and/or variable expressivity. The advent of next-generation sequencing, allowing the search for recurrent mutations or greater burden of rare variants in individual genes than expected by chance, can permit identification of such loci in the absence of classical segregation patterns. Very rare phenotypes, such as childhood PA, are promising candidates for such traits.

Using exome sequencing, we here identify five independent occurrences of the identical mutation in *CACNA1H* among 40 subjects with unexplained PA in childhood. *CACNA1H* encodes a voltage-gated calcium channel that is expressed in adrenal glomerulosa. Electrophysiology demonstrates that this variant causes reduced inactivation and a shift of activation to more hyperpolarized potentials, effects inferred to produce increased calcium influx and PA.

## Results

### Whole-exome sequencing of 40 subjects with PA

From a cohort of more than 1500 unrelated subjects referred for evaluation of genetic forms of hypertension, we identified 40 subjects diagnosed with hypertension and PA by age 10 years in whom disease-causing mutations in *CYP11B2*, *KCNJ5*, and *CACNA1D* (*Lifton et al., 1992*; *Choi et al., 2011*; *Scholl et al., 2013*) were excluded. Clinical details are shown in *Supplementary file 1A*. All subjects had hypertension with elevated aldosterone levels despite low plasma renin activity (PRA). None of the subjects studied were the offspring of consanguineous union. DNA from peripheral blood was subjected to exome capture and sequencing; mean coverage was 73 independent reads per targeted base (*Supplementary file 1B*). Variants were called as described in 'Materials and methods' (*Lemaire et al., 2013*).

We performed three analyses tailored to the expectation of a rare genetic disease ('Materials and methods'). We sought previously unreported (absent in dbSNP, NHLBI, 1000Genomes and Yale exome databases) protein-altering variants that occurred in more than one subject (*Supplementary file 1C*); we performed gene burden analyses to search for previously unreported or rare (minor allele frequency [MAF] < 0.01%) heterozygous variants that collectively occurred in any gene more often than expected by chance (*Supplementary file 1D*); we searched for rare (MAF < 0.1%) homozygous and potential compound heterozygous variants that collectively occurred in any genes more often than expected by chance (*Supplementary file 1E*).

### Identification of a recurrent novel variant in *CACNA1H*

There was only one result that surpassed genome-level significance: we found five apparently unrelated subjects with the identical previously unreported heterozygous A > G variant, resulting in a p.Met1549Val substitution in *CACNA1H*, which encodes the pore-forming alpha subunit of a T-type, low voltage-activated calcium channel (aka $Ca_V3.2$) (*Figure 1*, *Table 1*, *Supplementary file 1F*) (*Perez-Reyes, 2003*). This variant is absent among more than 129,000 alleles sequenced from diverse populations in the Exome Aggregation Consortium (*Exome Aggregation Consortium*), and Yale databases. No other *CACNA1H* alleles with allele frequencies <0.01% were found among our cohort. Like other $Ca^{2+}$ channel alpha subunits, CACNA1H contains four homologous repeats (I–IV), each with six transmembrane segments (S1–S6). The $CACNA1H^{M1549V}$ variant lies in the S6 segment of repeat III (*Marksteiner et al., 2001*). Sanger sequencing in each case confirmed the heterozygous variant (*Figure 1A*).

Three index cases were of European ancestry, one Hispanic, and one African American by self-report and principal component analysis (*Figure 1—figure supplement 1*). Members of the extended families were recruited, and sequencing of these subjects demonstrated that $CACNA1H^{M1549V}$ was a de novo mutation (absent in the biological parents) in both the index case of kindred 1347, and in the affected mother of the index case in kindred 1390 (*Figure 1*). Analysis of highly polymorphic markers confirmed paternity and maternity in both kindreds (*Supplementary file 1G*). This establishes independent occurrences of $CACNA1H^{M1549V}$ in these two kindreds. In the remaining three kindreds, the variant was transmitted to the index case from a parent, and samples from grandparents were not

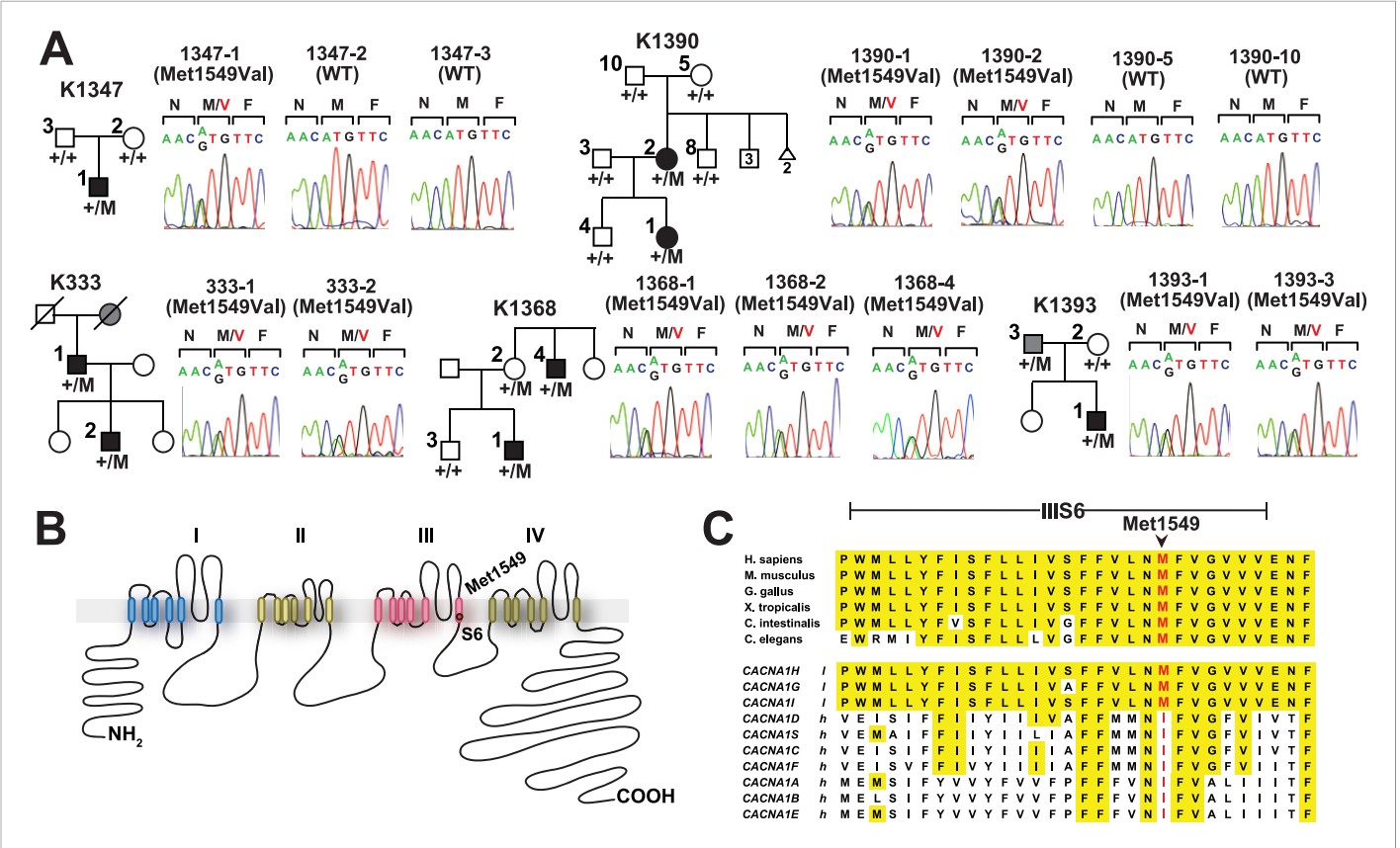

**Figure 1**. Kindreds with hypertension and primary aldosteronism (PA) with *CACNA1H*[M1549V] mutation at conserved position of S6 domain. (**A**) Pedigrees of kindreds with *CACNA1H*[M1549V] mutation are shown. Studied subjects with early-onset hypertension are shown as black filled symbols, and subjects with early-onset hypertension by family history (K333) or low renin with normal blood pressure (K1393) are shown as grey filled symbols. Genotypes are indicated below each symbol (+/+ denotes wild type sequence; +/M denotes heterozygosity for *CACNA1H*[M1549V] variant). Corresponding Sanger sequencing results for selected subjects are depicted to the right. (**B**) Transmembrane structure of Ca$_V$3.2 (encoded by *CACNA1H*), the pore-forming subunit of a voltage-gated Ca$^{2+}$ channel, is shown. These channels have four internal homologous repeats (I–IV), each with six transmembrane segments (S1–S6) and a membrane-associated loop between the pore-forming S5 and S6 segments. The p.Met1549Val mutation is located in S6 of repeat III. (**C**) Conservation of CACNA1H[M1549] in CACNA1H orthologs and paralogs. The amino acid sequences of the S6 segment of domain III of CACNA1H, orthologs and paralogs are shown. The S6 segment, including Met1549, is virtually completely conserved (highlighted in yellow) among orthologs and all paralogs that are activated by small changes in membrane potential (l, low voltage-activated) but not those activated by large changes (h, high voltage-activated). M1549 is part of the Met-Phe-Val sequence that is implicated in rapid channel inactivation (*Marksteiner et al., 2001*).

The following source data and figure supplements are available for figure 1:

**Source data 1**. Source data corresponding to *Figure 1*.

**Figure supplement 1**. Cohort population structure by principal component analysis (PCA).

available for further analysis of transmission (*Figure 1*). Analysis of kinship coefficients using SNP genotypes of affected subjects from Illumina Human 1M-Quad beadchips and the KING algorithm (*Manichaikul et al., 2010*) provided no evidence that these three kindreds shared recent common ancestry (*Supplementary file 1H*, 'Materials and methods'). Further, haplotypes flanking the *CACNA1H* mutation were phased using the BEAGLE program, revealing that the maximum shared haplotype flanking the *CACNA1H* mutation among these three kindreds was only 53.6 kb (87.0 kb for the two European kindreds, *Figure 2*). From this data, the maximum likelihood estimate of the number of generations since the last shared common ancestor among these subjects is estimated to be 714 generations 95% CI 290-1268 (*Genin et al., 2004*). A more conservative analysis identifying homozygous discordant SNPs (eliminating inference of haplotypes by phasing) still limited the

Table 1. Clinical features of index cases with CACNA1H$^{M1549V}$

| Subject ID | Gender | Age dx | BP (%ile) | Aldo (ng/dl) | PRA (ng/ml/hr) | ARR (ng/dl: ng/ml/hr) |
|---|---|---|---|---|---|---|
| 1347-1 | M | 3 yrs | 160/105 (>99th) | 20 | <0.1 | >200 |
| 1390-1 | F | 7 yrs | 150/90 (>99th) | 66 | 0.2 | 330 |
| 1368-1 | M | 8 yrs | 140/90 (>99th) | 20 | <0.2 | >100 |
| 333-2 | M | 9 yrs | 192/144 (>99th) | 40 | <0.7 | >57 |
| 1393-1 | M | 2 mos | 170/110 (>99th) | 87 | <0.6 | >145 |

M, male; F, female; age dx, age at diagnosis of hypertension; yrs, years; mos, months; BP, blood pressure; (%ile), percentile adjusted for age and gender; Aldo, serum aldosterone; PRA, plasma renin activity; ARR, aldosterone:renin ratio, values >20 with aldosterone level >15 are considered indicative of primary aldosteronism (PA).

shared haplotype to less than 127.1 kb, consistent with the results of phasing using BEAGLE. These findings indicate that the CACNA1H$^{M1549V}$ mutation in these three kindreds has not been inherited from a recent common ancestor, and has either arisen independently or has been inherited from an extremely remote common ancestor. The latter possibility is extremely unlikely given the absence of this mutation in more than 129,000 alleles studied to date.

The probability of finding any recurrent protein-altering de novo mutation among 40 kindreds is ~$4.0 \times 10^{-6}$ (see 'Materials and methods'). Even with a conservative estimate of the allele frequency of CACNA1H$^{M1549V}$ of 0.00001 in the general population ('conservative' because it has never been seen among >129,000 alleles in subjects not selected for early PA), the probability of finding three additional instances of this mutation in 38 unrelated subjects is ~$8.4 \times 10^{-12}$. Combined, the probability of finding these five instances of the identical variant by chance is conservatively estimated to be $3.4 \times 10^{-17}$, providing extremely strong statistical support for the role of this mutation in PA.

## CACNA1H is expressed in human adrenal glomerulosa

If the CACNA1H$^{M1549V}$ mutation causes early-onset PA, CACNA1H (Ca$_V$3.2) should be expressed in human adrenal glomerulosa. CACNA1H transcripts have previously been reported in human kidney, liver, heart and brain (Cribbs et al., 1998), and our prior analysis of gene expression of human adrenal cortex showed that CACNA1H was the second most highly expressed calcium channel alpha subunit, after CACNA1D (Scholl et al., 2013). We performed immunohistochemistry with two different antibodies specific for the encoded channel protein (Ca$_V$3.2), demonstrating strong staining of human adrenal glomerulosa; this staining was abolished after preincubation with immunogenic peptide (Figure 3). These results are consistent with prior in situ hybridization and electrophysiological studies of rodent and bovine glomerulosa (Schrier et al., 2001; Hu et al., 2012) as well as a recent study of human adrenal gland (Felizola et al., 2014).

## Clinical features of subjects with CACNA1H$^{M1549V}$ variant

The clinical features of the index cases harboring the CACNA1H$^{M1549V}$ variant were uniform. All presented with hypertension by age 10 and had persistent inappropriate elevation of serum aldosterone with suppressed PRA and high aldosterone:PRA ratio, indicative of autonomous adrenal aldosterone production (Table 1). Adrenal imaging by computed tomography, magnetic resonance or ultrasound showed no evidence of mass or hyperplasia at the time of presentation. There were no other recurrent or distinctive features of the index cases, specifically no history of seizures, neurologic or neuromuscular disorders as found in patients with CACNA1D mutations (Scholl et al., 2013). Additional details are presented in Appendix 1.

By direct Sanger sequencing, we identified five additional CACNA1H$^{M1549V}$ mutation carriers among family members, including four parents and one uncle of an index case (Figure 1). Of these five, three were diagnosed with early severe hypertension while two were not, and in fact were normotensive as adults (Appendix 1 and Supplementary file 1I). For example, subject 1390-2 was diagnosed with severe hypertension and PA at age 17; her hypertension was difficult to control, leading to unilateral adrenalectomy at age 29. Her hypertension nonetheless recurred, requiring reinstitution of treatment.

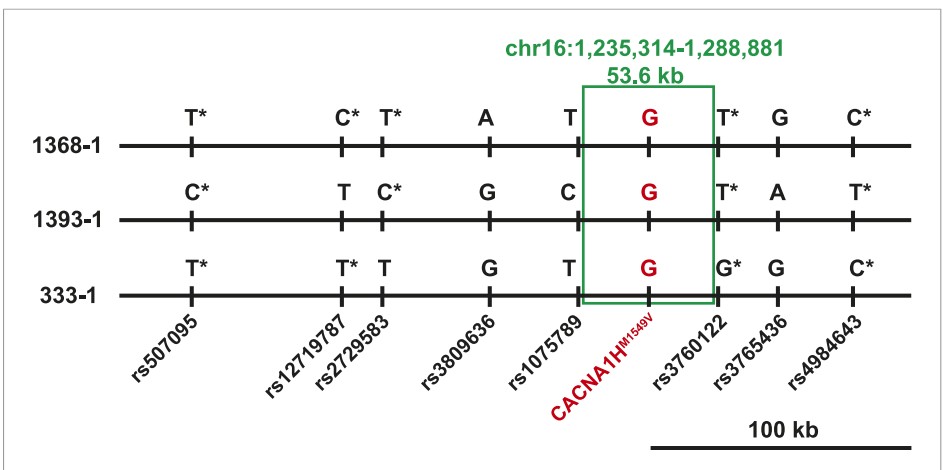

**Figure 2**. Shared haplotypes in subjects with inherited *CACNA1H*<sup>M1549V</sup> variant. Haplotypes of three affected individuals from kindreds without proven de novo occurrence of *CACNA1H*<sup>M1549V</sup> variant were phased using BEAGLE ('Materials and methods') (*Browning and Browning, 2007*). This analysis identified a very small maximum interval shared among all three individuals (~53.6 kb, green box) flanked by rs1075789 and rs3760122. If only homozygous discordant calls (*) were considered in the absence of phasing, the maximum interval shared by all three subjects would be 127.1 kb and the longest pairwise shared haplotype would be 200.0 kb between 1393-1 and 333-1.

Interestingly, the histology of her adrenal gland showed striking microscopic hyperplasia. While the normal adrenal glomerulosa comprises only a few cell layers and is about 70 μm in depth, the glomerulosa of subject 1390-2 was ~30 cell layers and ~300 μm in depth (*Figure 4A–C*). CACNA1H was expressed in the hyperplastic glomerulosa layer (*Figure 4*, Appendix 1). In this kindred, the *CACNA1H*<sup>M1549V</sup> variant arose concordantly and segregated precisely with PA and early hypertension (*Table 2*). Two mutation carriers (1368-2 and 1393-3) were normotensive as adults and had not been diagnosed as hypertensive in childhood; in subject 1393-3, PRA was at the lower limit of normal with normal aldosterone level, while PRA and aldosterone levels were normal in 1368-2 (*Supplementary file 1I*).

## Specificity of *CACNA1H*<sup>M1549V</sup> variant for early-onset PA

To explore the specificity of this mutation for early-onset PA, we performed targeted Sanger sequencing for the *CACNA1H*<sup>M1549V</sup> variant in germline DNA of 1632 additional unrelated subjects, comprising 324 subjects with PA diagnosed after age 10 years, 96 with hypertension and bilateral adrenal hyperplasia, and 1212 referred for potential genetic causes of hypertension without evidence of PA. We also sequenced tumor DNA of 90 APAs, including 40 that did not have mutations in previously implicated genes (*KCNJ5*, *CACNA1D*, *ATP1A1*, *ATP2B3*, and *CTNNB1* [*Choi et al., 2011*; *Azizan et al., 2013*; *Beuschlein et al., 2013*; *Scholl et al., 2013*]). No additional *CACNA1H*<sup>M1549V</sup> mutations were identified, demonstrating striking specificity for early-onset PA.

## *CACNA1H*<sup>M1549</sup> lies in a conserved MFV motif

Members of the Ca$_V$3 family are activated by small depolarizing changes in the membrane potential (activation threshold ~ −60 mV) and display very fast voltage-dependent inactivation (*Perez-Reyes, 2003*). Methionine at the position corresponding to CACNA1H<sup>M1549</sup> is conserved in the S6 helix of repeat three in all identified orthologs, including invertebrates. In addition, methionine occurs at the paralogous position in other calcium channels activated by small depolarizing potential changes (*Figure 1C*). Prior studies of Ca$_V$3.1 (CACNA1G) have shown that methionine 1549 lies in a methionine-phenylalanine-valine (MFV) tripeptide that regulates channel inactivation (*Hering et al., 1998*; *Marksteiner et al., 2001*). Mutation of the homologous methionine in Ca$_V$3.1 to isoleucine or alanine results in delayed channel inactivation (*Marksteiner et al., 2001*), and related calcium channels with isoleucine at the homologous position inactivate more slowly than those with methionine (*Hering et al., 1997*) (*Figure 1C*).

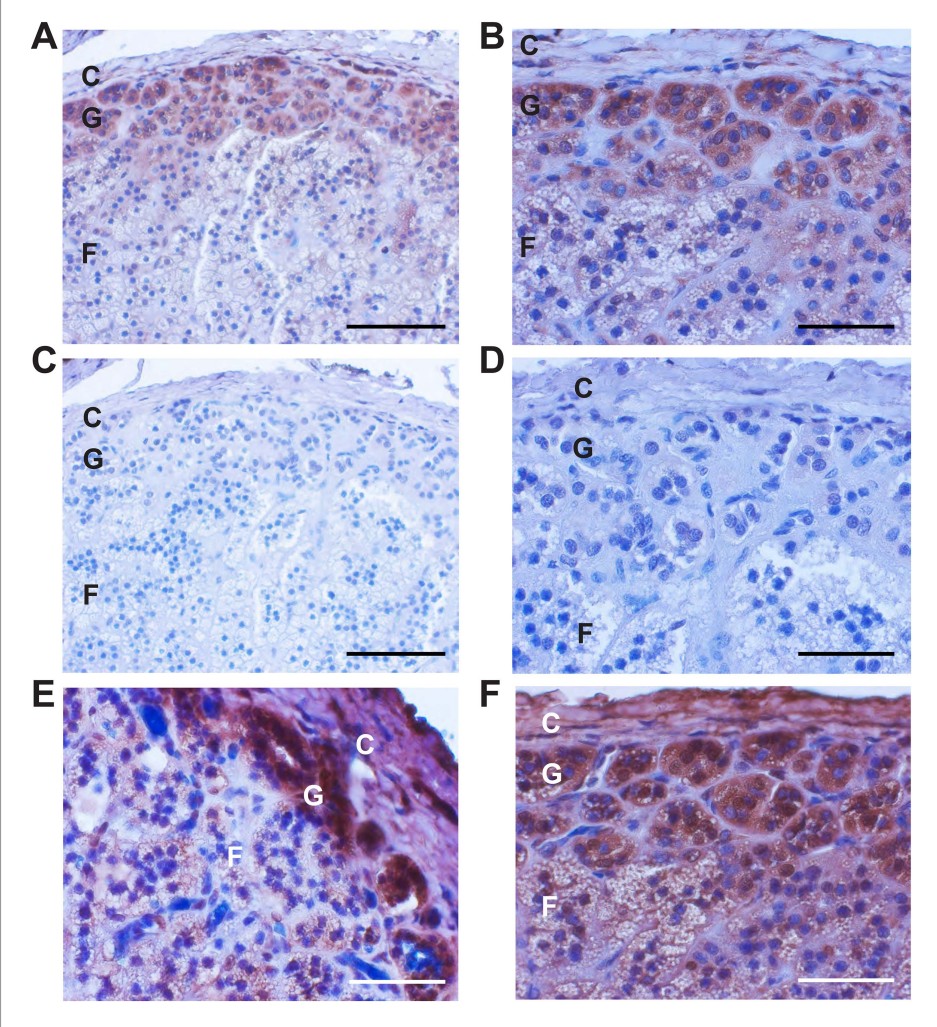

**Figure 3**. Immunohistochemistry of Ca$_V$3.2 in normal human adrenal gland. Sections of normal human adrenal are shown. C denotes adrenal capsule; G, glomerulosa; F, fasciculata. (**A**) Normal adrenal gland stained with hematoxylin and an antibody to Ca$_V$3.2 (Alomone). (**B**) Higher power image of adrenal in panel (**A**). (**C**, **D**) Absence of staining after preincubation of the antibody with the antigenic peptide, demonstrating specificity. (**E**) A second normal human adrenal gland stained for CACNA1H as in (**A**, **B**). (**F**) Gland from (**A**–**D**) stained with a second α-CACNA1H antibody (Santa Cruz). Scale bars, 100 μm (**A**, **C**); 50 μm (**B**, **D**, **E**, **F**). The results demonstrate expression of Ca$_V$3.2 in the normal zona glomerulosa, which is only several cells in depth.

## CACNA1H$^{M1549V}$ causes loss of normal inactivation

To assess the biophysical properties of CACNA1H$^{M1549V}$, we heterologously expressed either CACNA1H$^{WT}$ or CACNA1H$^{M1549V}$ in HEK293T cells and performed whole-cell patch clamp recordings (*Figure 5A*). Upon depolarizing voltage steps from −90 mV, CACNA1H$^{WT}$ showed fast activation of calcium currents followed by rapid inactivation, consistent with prior studies (*Cribbs et al., 1998*). In contrast, CACNA1H$^{M1549V}$ exhibited marginally slower activation followed by a dramatically slowed inactivation. While CACNA1H$^{WT}$ is virtually fully inactivated by 400 ms, CACNA1H$^{M1549V}$ shows strong tail currents after returning to the holding potential of −90 mV (*Figure 5A*), demonstrating loss of normal inactivation, an effect still evident after sustained depolarization for 5 s (*Figure 5B*).

We fitted mono-exponential equations to the decay phase of the calcium current between −50 mV and +30 mV. The determined time constants represent the mean time at which the current has decreased to 1/e of its initial amplitude; the results demonstrate ~10-fold slower inactivation of CACNA1H$^{M1549V}$ compared to CACNA1H$^{WT}$ (p < 0.001 at all voltages studied, *Figure 5C*). In contrast,

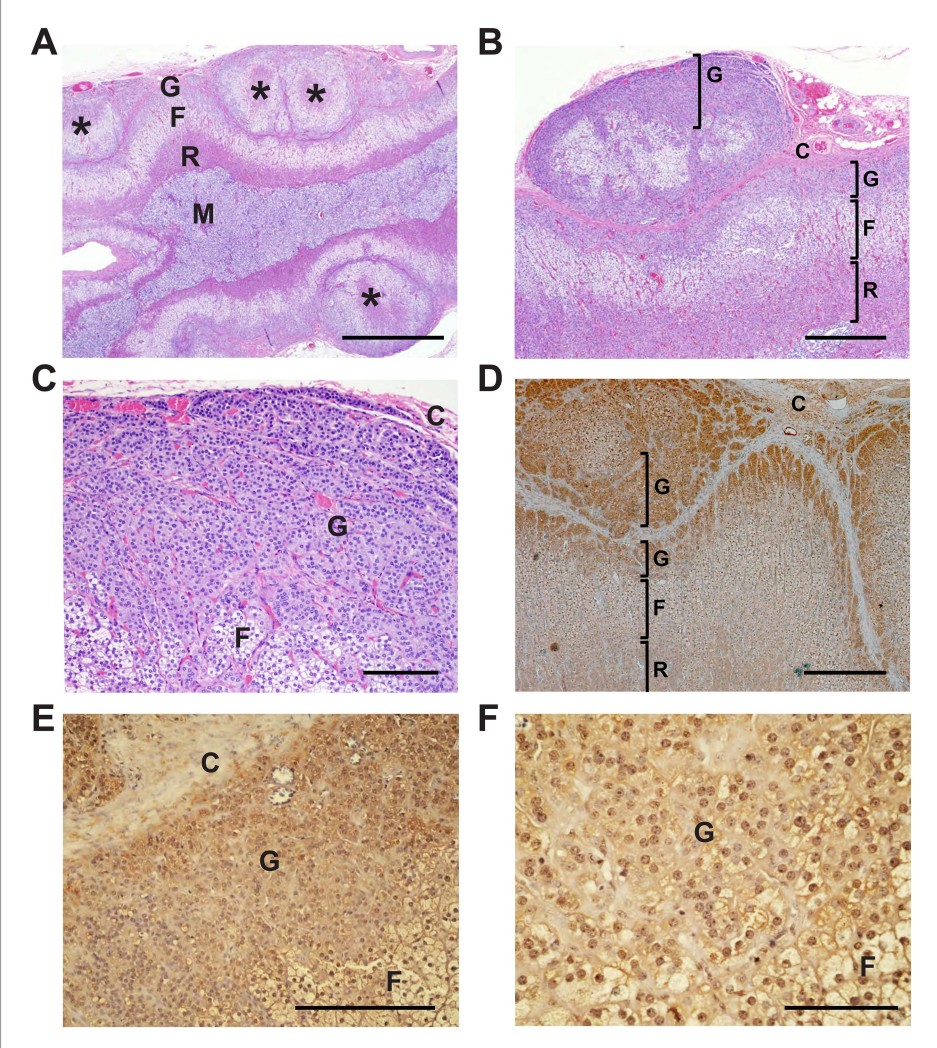

**Figure 4**. Glomerulosa hyperplasia in adrenal gland of subject 1390-2 with *CACNA1H*[M1549V] mutation. C denotes adrenal capsule; G, glomerulosa; F, fasciculata; R, reticularis; M, medulla. (**A**) Low power image stained with hematoxylin and eosin. Scale bar 1000 μm. (**B, C**) Higher power images of adrenal from panel (**A**), scale bars 400 μm (**B**) or 100 μm (**C**). The mutant adrenal shows marked zona glomerulosa hyperplasia, with micronodular invasion of the capsule (denoted by *). (**D**) Same adrenal gland stained with hematoxylin and antibody to $Ca_V3.2$ (Santa Cruz), demonstrating specific staining of zona glomerulosa. Scale bar, 400 μm. (**E, F**), higher power images stained with second antibody to $Ca_V3.2$ (Alomone). Scale bars 250 μm (**E**) or 100 μm (**F**). $Ca_V3.2$ is expressed in the hyperplastic zona glomerulosa.

activation and recovery from inactivation were only marginally slower in mutant channels (*Figure 5D*, *Figure 5—figure supplement 1*).

## CACNA1H[M1549V] leads to a shift of activation to less depolarizing potentials

We also observed a significant shift of activation to less depolarizing potentials (*Figure 6*). CACNA1H[WT] showed half-maximal activation ($V_{1/2}$) at −38.9 ± 1.1 mV (N = 11); in contrast, CACNA1H[M1549V] showed $V_{1/2}$ of −44.2 ± 1.1 mV (N = 11, p = 0.003), resulting in a lower threshold for activation and increase in size of the 'window current', the area under the intersection of activation and inactivation curves where a fraction of channels are constitutively open. There was no significant effect on single channel conductance (*Figure 6—figure supplement 1*).

**Table 2**. Laboratory values of carriers and non-carriers of *CACNA1H*[M1549V] in kindred 1390

| Subject ID | Gender | Age (yrs) | K+ (mmol/l) | Aldo (ng/dl) | PRA (ng/ml/hr) | ARR (Aldo/PRA) | Direct renin (μIU/ml) | Aldo/direct renin |
|---|---|---|---|---|---|---|---|---|
| Carriers | | | | | | | | |
| 1390-1 | F | 15 | 3.7 | 37 | 0.42 | 88.1 | NA | NA |
| 1390-2 | F | 29 | 3.5 | 22 | NA | NA | 3 | 7.3 |
| Non-Carriers | | | | | | | | |
| 1390-4 | M | 17 | 4.3 | 2 | 1.65 | 1.2 | NA | NA |
| 1390-8 | M | 37 | 3.8 | <1 | 2.25 | <0.4 | NA | NA |
| 1390-5 | F | 62 | 4.1 | 16 | 18.92 | 0.8 | NA | NA |
| 1390-6 | F | 51 | 4.0 | 4 | 1.87 | 2.1 | NA | NA |
| 1390-7 | F | 46 | 4.3 | 3 | 1.67 | 1.8 | NA | NA |

M, male; F, female; Age (yrs), Age in years when sample was obtained; K+, serum potassium (reference 3.5–5.5 mmol/l); Aldo, serum aldosterone; PRA; direct renin <5 is indicative of volume-mediated hypertension; ARR, aldosterone:renin ratio, values >20 (using PRA) or values of aldo/direct renin >2.4 with aldosterone level greater than 15 are considered indicative of PA. Blood samples were drawn on the same day, and values were determined in the same laboratory (except for 1390-2, in whom values are pre-adrenalectomy at age 29). See *Figure 1* for relationships. 1390-6 and −7 are not included in *Figure 1*, and are sisters of 1390-5.

Collectively, the changes in inactivation and voltage-dependence of activation cause $Ca^{2+}$ influx at membrane potentials close to the resting potential and result in channels that remain open longer, allowing increased $Ca^{2+}$ entry.

## Discussion

These findings define a previously unrecognized form of PA resulting from a recurrent germline gain of function mutation in the $Ca^{2+}$ channel encoded by *CACNA1H*. The extremely strong statistical evidence implicating this mutation, its clear gain of function effect, and the known role of $Ca^{2+}$ signaling on aldosterone production and cell proliferation (*Spät and Hunyady, 2004*) all strongly support this conclusion. The effects of this mutation phenocopy the adrenal effects of PA-causing mutations in *KCNJ5* (*Choi et al., 2011*) and *CACNA1D* (*Scholl et al., 2013*), demonstrating a shared final common pathway by which PA results from increased $Ca^{2+}$ entry via voltage-gated channels. These results allow a simple genetic test for this specific cause of severe hypertension and suggest that inhibition of mutant CACNA1H activity would ameliorate hypertension in patients with this mutation. While the *CACNA1H*[M1549V] mutation explains a significant fraction of the early PA cases, the causes of the remaining cases in our cohort are still unknown.

There is striking genotype–phenotype correlation among patients with germline and somatic mutations in *KCNJ5* and *CACNA1H*. Several recurrent germline mutations in *KCNJ5* (e.g., p.Gly151Arg and p.Thr158Ala) support robust cell proliferation leading to massive adrenal hyperplasia identifiable on CT scan, leading to adrenalectomy (*Choi et al., 2011*; *Scholl et al., 2012*). In contrast, another recurrent mutation in *KCNJ5* (p.Gly151Glu) shows no or minimal hyperplasia discernable by adrenal imaging (*Mulatero et al., 2012*; *Scholl et al., 2012*). While the former mutations are also found as somatic mutations in about 40% of APAs (*Choi et al., 2011*; *Mulatero et al., 2012*), the latter have not been found in more than 900 APAs (*Scholl and Lifton, 2013*). This phenotypic difference is likely accounted for by different effects on $Na^+$ conductance- the germline mutations that are not associated with APAs or hyperplasia cause markedly greater $Na^+$ conductance, resulting in very high cell lethality, preventing sustained increases in cell mass (*Mulatero et al., 2012*; *Scholl et al., 2012*). Similarly, adrenal glands with *CACNA1H*[M1549V] show little or no hyperplasia by CT scan and neither this mutation nor other activating mutations in *CACNA1H* have been seen in APAs. We have not observed high cell lethality in HEK293T cells expressing CACNA1H[M1549V]. Germline mutations that cause massive hyperplasia and somatic mutations that cause APA formation likely require an optimal $Ca^{2+}$ signal, one that is high enough to support proliferation but not so high as to cause cell lethality (*Berridge et al., 2000*).

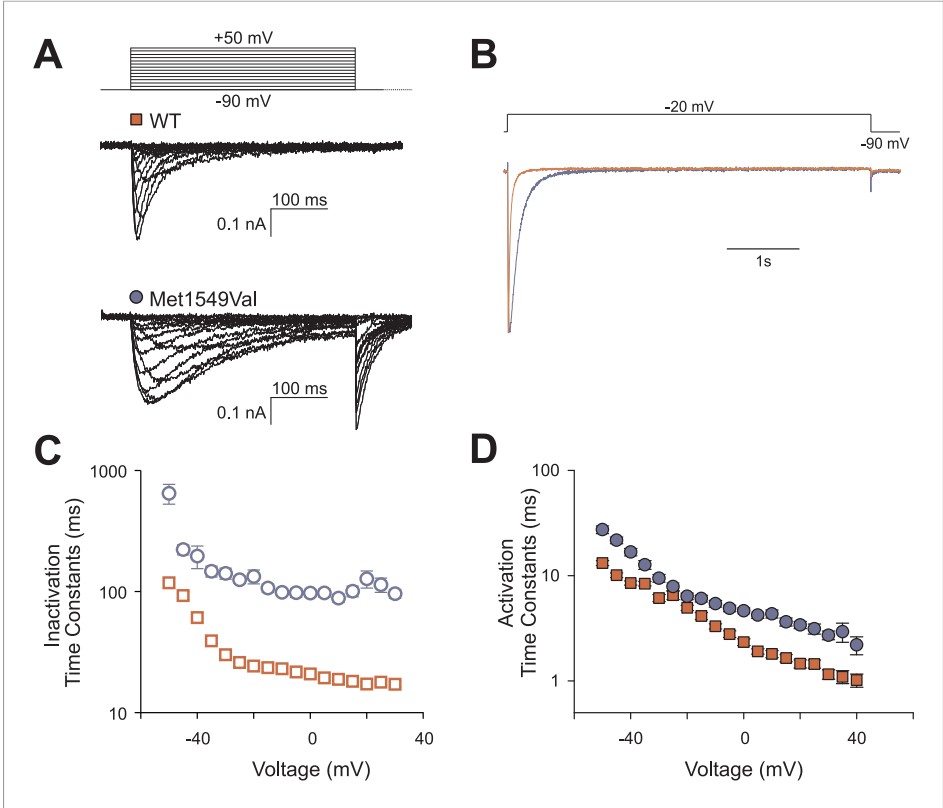

**Figure 5**. CACNA1H[M1549V] impairs channel inactivation. Whole-cell patch clamp recordings were performed in HEK293T cells transfected with *CACNA1H[WT]* or *CACNA1H[M1549V]*. (**A**) Cells were held at −90 mV, and voltage steps between −90 and +50 mV were applied to elicit calcium currents, followed by a step to −90 mV to evoke tail currents. Representative recordings show rapid activation and inactivation of CACNA1H[WT] currents and delayed inactivation of CACNA1H[M1549V]. Tail currents are exclusively present in CACNA1H[M1549V] and suggest the presence of non-inactivated mutant channels at the end of the depolarizing pulse. (**B**) Tail currents are still present after a 5-s pulse to −20 mV. The fraction of non-inactivated channels after 5 s was determined by dividing the peak amplitude at −20 mV before and after 5 s long pulses to voltages between −90 and −20 mV in 5 mV increments (CACNA1H[M1549V]: 6.7 ± 1%, N = 12; CACNA1H[WT]: 2.4 ± 0.5%, N = 9; p = 0.004, protocol not shown in figure). (**C**) Exponential fits of the current decay between −50 and +30 mV provide inactivation time constants. Data from CACNA1H[M1549V] are shown in blue circles, CACNA1H[WT] data are shown in red squares. The mutant channel shows almost 10-fold slower inactivation than wild-type (N = 9 for CACNA1H[WT], N = 7–14 for CACNA1H[M1549V], p < 0.001 across all voltages studied, Mann–Whitney rank sum test). (**D**) In contrast, activation time constants at different voltages are only slightly slower in CACNA1H[M1549V] compared to [WT] (cf. 'Materials and methods' for details). Source files are available in *Figure 5—source data 1*.

The following source data and figure supplements are available for figure 5:

**Source data 1**. Source data corresponding to *Figure 5*.

**Figure supplement 1**. Recovery from inactivation is slightly slower in CACNA1H[M1549V].

While germline mutations in *KCNJ5* and *CACNA1D* were discovered following the initial identification of their somatic mutations in APAs (*Choi et al., 2011*; *Scholl et al., 2013*), the discovery of the recurrent *CACNA1H* mutation relied entirely on brute force sequencing of patients with early severe aldosteronism and hypertension. The occurrence of de novo mutations, the reduced penetrance in adults with the absence of large multiplex families, as well as the absence of distinctive phenotypes that distinguish these patients from others with early hypertension and aldosteronism, all suggest reasons that *CACNA1H* mutations were not previously linked to PA.

The reduced penetrance in adults in particular is interesting—two mutation-carrier parents were normotensive as adults, without clear evidence of PA. Incomplete penetrance among some carriers of

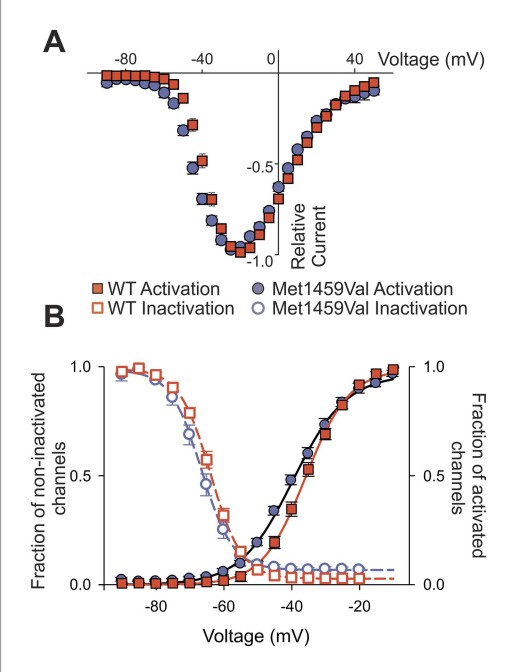

**Figure 6**. CACNA1H[M1549V] shifts activation to more hyperpolarized potentials. (**A**) Current-voltage plots and (**B**) activation curves show a shift of $V_{1/2}$ for activation of the mutant channel to less depolarizing potentials. The voltage dependence of inactivation is shown as open circles or squares. For CACNA1H[M1549V], the area under the intersection of activation and inactivation curves (where a fraction of channels show continuous activity) is larger and shifted to more hyperpolarized potentials compared to CACNA1H[WT], allowing for increased constitutive $Ca^{2+}$ influx at potentials close to the resting potential of zona glomerulosa cells (*Hu et al., 2012*). Source files are available in *Figure 6—source data 1*.

The following source data and figure supplements are available for figure 6:

**Source data 1**. Source data corresponding to *Figure 6*.

**Figure supplement 1**. CACNA1H[M1549V] and CACNA1H[WT] whole-cell current densities and non-stationary noise analysis.

mutations that cause aldosteronism (*Stowasser et al., 1995*; *Mulatero et al., 2002*; *Scholl et al., 2013*) has been previously described. The explanations for these effects remain unclear, however age-dependent activity of the renin-angiotensin system and the ability of older individuals to modulate dietary salt intake in response to physiologic demand are potential contributors. This is well described in the case of heterozygous loss of function mutation in the receptor for aldosterone (the mineralocorticoid receptor, MR). These patients have life-threatening salt-wasting and volume depletion in the first years of life due to low signaling through MR, but are asymptomatic as adults. Adult subjects show increased dietary salt intake and increase MR signaling by induction of the renin-angiotensin system, thereby markedly increasing aldosterone levels (*Geller et al., 1998*). Other possible mechanisms for incomplete penetrance include genetic modifiers either in cis or in trans, including the possibility of somatic mosaicism resulting in absence of the gain of function mutation the adrenal gland (*Youssoufian and Pyeritz, 2002*). While such mosaicism cannot be excluded, Sanger sequence traces provided no suggestion of mosaicism in circulating white blood cell or saliva DNA (*Figure 1A*).

CACNA1H[M1549V] shows constitutive activity at membrane potentials close to the resting potential, allowing channels to be activated despite suppression of the renin-angiotensin system and absence of hyperkalemia. CACNA1H[M1549V] channels also show strikingly delayed inactivation, a finding similar to mechanisms in several other channelopathies (*Cannon et al., 1991*; *Lerche et al., 1993*; *Scholl et al., 2013*). In glomerulosa, delayed inactivation is inferred to increase the period of membrane potential depolarizations. Interestingly, recent studies in mouse have implicated CACNA1H activity in regular glomerulosa membrane potential oscillations that may amplify small changes in membrane potential to produce significant $Ca^{2+}$

signals (*Hu et al., 2012*). Thus this regular activation of CACNA1H, together with a shift of activation to less depolarized potentials and prolonged activity, provides a mechanism for increased $Ca^{2+}$ entry, leading to aldosteronism. While a common variant in *CACNA1H* has been suggested to be associated with blood pressure in a small genome-wide association study of African American individuals (*Adeyemo et al., 2009*), this result did not pass criteria for genome-wide significance, was only found after exclusion of hypertensive individuals, and was not replicated in larger studies (*International Consortium for Blood Pressure Genome-Wide Association Studies et al., 2011*; *Kidambi et al., 2012*).

The apparent limitation of the phenotype associated with *CACNA1H[M1549V]* to PA with hypertension despite the expression of *CACNA1H* in other organs including heart and brain (*Cribbs et al., 1998*) is notable, and underscores the challenges in predicting human phenotypes from knowledge of underlying mutations. No mutation carrier had a history of seizures or cardiac

arrhythmia. While some prior studies have suggested a role of rare gain of function mutations in *CACNA1H* in epilepsy (*Chen et al., 2003*; *Liang et al., 2006*; *Heron et al., 2007*), these studies have not approached genome-wide levels of significance, do not appear to confer high risk, and have not been uniformly replicated (*Heron et al., 2004*; *Chioza et al., 2006*; *Liang et al., 2006*).

Our findings are consistent with evidence supporting a normal role for CACNA1H in the regulation of human aldosterone biosynthesis (*Felizola et al., 2014*). Because CACNA1H is activated by small depolarizing changes in glomerulosa membrane potential, it is likely activated in response to small day-to-day changes in serum $K^+$ concentration and angiotensin II levels that require fine adjustments in aldosterone production to maintain volume and electrolyte balance. In contrast, CACNA1D, which is the most highly expressed calcium channel in adrenal cortex, and which shows larger single channel conductance than CACNA1H (*Michels et al., 2002*; *Bock et al., 2011*), is only activated by large depolarizations. Activation of this channel likely contributes to the high levels of aldosterone produced in response to marked volume depletion or hyperkalemia. We suggest that CACNA1H and CACNA1D act in series in the regulation of aldosterone, with CACNA1H being activated in response to small, frequent physiologic perturbations and CACNA1D in response to more infrequent large physiologic challenges.

These findings also raise the question whether inhibition of wild type CACNA1H would lower blood pressure or aldosterone production. In the general population, loss-of-function variants in *CACNA1H* are very rare (cumulative frequency of splice site, frameshift and nonsense variants in the ExAC database of 0.06%, resulting in expected compound heterozygosity or homozygosity in about 1 in 2.6 million subjects), making such studies challenging. It seems plausible that loss of CACNA1H could be compensated by activation of the renin-angiotensin system, leading to greater glomerulosa cell depolarization with consequent activation of CACNA1D, maintaining normal aldosterone production and blood pressure. Consistent with this suggestion, blood pressure was reportedly unchanged in a *CACNA1H* knockout mouse model, although aldosterone levels were not reported (*Chiang et al., 2009*). Similarly, selective inhibitors of CACNA1H inhibit aldosterone production in vitro (*Rossier et al., 1998*; *Perez-Reyes et al., 2009*), but do not apparently reduce aldosterone levels or blood pressure in vivo (*Schmitt et al., 1992*; *Ragueneau et al., 2001*). Whether additional non-dihydropyridine compounds will prove to be more effective in lowering aldosterone levels or blood pressure will be interesting to assess.

## Materials and methods

### Subjects
PA was diagnosed based on elevated ARR (>20 ng/dl:ng/ml/hr), typically with aldosterone >15 ng/dl, or marginally elevated values in the presence of unexplained hypokalemia (*Funder et al., 2008*). Venous blood or saliva samples were obtained from subjects with unexplained early-onset PA and family members. Research protocols were approved by the local institutional review board (IRB), and informed consent was obtained from all research participants.

### DNA preparation, and exome sequencing
DNA was prepared from venous blood or saliva samples using standard procedures. Exome capture was performed using the 2.1M NimbleGen Exome reagent (Roche NimbleGen, Madison, WI), and 75 base paired end sequencing on the Illumina (San Diego, CA) platform was performed as previously described (*Lemaire et al., 2013*). Coverage statistics are provided in *Supplementary file 1B*.

### Sanger sequencing of genomic DNA and genotyping of parent-offspring trios
Direct bidirectional Sanger sequencing of *CACNA1H^P1523-R1584* from genomic DNA of indicated subjects was performed following PCR amplification using primers

CACNA1H_25F (5′-GACCCACCGCCTCTGTG-3′) and CACNA1H_25R (5′-AGCGCCT-TACTCCTGCG-3′).

Parent-offspring trios were genotyped as previously described, except for locus D7S820 in kindred 1390 (primers [5′-ATGTTGGTCAGGCTGACTATG-3′] and [5′-GATTCCACATTTATCCTCATTGAC-3′])

(*Scholl et al., 2013*). Alleles without known frequencies in the population were omitted from the analysis.

## Immunohistochemistry

Normal human adrenal tissue was obtained from the Yale Pathology archive, and adrenal tissue from subject 1390-2 from Pathology Services of Beaufort/Charleston (South Carolina, USA). Immunohistochemistry was performed as previously described (*Scholl et al., 2013*). Primary antibodies were α-Ca$_V$3.2 (#ACC-025, Alomone, Jerusalem, Israel) or T-type Ca++ CP α1H (SC-25691, Santa Cruz Biotechnology, Santa Cruz, CA), both at dilutions of 1:100; secondary antibody was donkey α-rabbit (#035-152, 1:500, Jackson, Bar Harbor, ME). For the Alomone antibody, preincubation with the antigenic peptide (1:1, wt/wt in 10% FBS) was performed for 1 hr at RT. Both antibodies were tested on two independent glands. H&E staining was performed at Yale Research Histology using routine procedures.

## Molecular cloning

Myc-DDK-tagged CACNA1H in pCMV6-Entry was obtained from Origene (Rockville, MD) (RC212772, NM_021098.2). Site-directed mutagenesis (QuikChange, Agilent Technologies, Santa Clara, CA) was performed to introduce the p.Met1549Val mutation according to the manufacturer's instruction. Each construct was validated by sequencing of the entire coding region.

## Transient transfection and electrophysiological recordings

Culturing of HEK293T cells was performed as described (*Scholl et al., 2013*). Cells were transfected with 3 μg of CACNA1H$^{WT}$ or CACNA1H$^{M1549V}$ expression plasmids. For each construct, two clones were functionally tested. Whole cell patch clamp recordings were performed on a HEKA EPC10 amplifier (HEKA Elektronik, Ludwigshafen, Germany) as described previously (*Scholl et al., 2013*). The extracellular solution contained: 5 mM CaCl$_2$, 125 mM TEA-Cl, 10 mM HEPES, 15 mM Mannitol, pH 7.4. Pipette solution contained: 100 mM CsCl, 5 mM TEA-Cl, 3.6 mM PCr-Na$_2$, 10 mM EGTA, 5 mM Mg-ATP, 0.2 mM Na-GTP, 10 mM HEPES, pH 7.4 (titration with CsOH).

Voltage dependences of activation were determined from the peak current–voltage relation and fit by a Boltzmann function as described (*Marcantoni et al., 2010*; *Scholl et al., 2013*). The fraction of non-inactivated channels was determined by dividing the peak amplitude at −20 mV before and after 5 s long pulses to voltages between −90 and −20 mV. Time courses of activation or inactivation were analyzed by fitting a mono-exponential function (*Scholl et al., 2013*). The recovery from inactivation was measured using envelope protocols consisting of an inactivation of channels during a 5 s pulse to −20 mV followed by holding the membrane potential at −90 for increasing durations (*Coulter et al., 1989*). Afterwards, peak currents at −20 mV were measured and divided by the previous peak current. A plot of these ratios vs the duration of the pulse to −90 mV was fit with a mono-exponential function to obtain time constants for the recovery from inactivation.

Non-stationary noise analysis was performed as described (*Hebeisen and Fahlke, 2005*) using a voltage protocol that activates channels at −20 mV followed by the analysis of the decay of currents and variance at −90 mV. The initial variance at the holding potential of −90 mV before activation was regarded as background variance and subtracted from the recordings. The Lorentzian noise produced by channel opening and closing depends on the unitary current amplitude (i), the number of channels (N), and the absolute open probability (P):

$$\sigma^2 = N \cdot i^2 \cdot p \cdot (1-p). \tag{1}$$

Since the macroscopic current amplitude is given by

$$I = N \cdot p \cdot i, \tag{2}$$

the variance-current relationship results in a quadratic distribution:

$$\sigma^2 = i \cdot \langle I \rangle - \left( \frac{\langle I \rangle^2}{N} \right). \tag{3}$$

The single channel amplitude (i) was derived from the initial slope of a plot of the variance against the mean isochronal current results. Due to a low open probability ($p < 0.5$) at 5 mM of external $Ca^{2+}$, the recorded data points only described a small part of the usual parabola and did not allow for determination of the number of channels and open probabilities.

Data were analyzed in FitMaster (HEKA Elektronik), SigmaPlot (Jandel Scientific, San Rafael, CA) and Python. Statistical comparisons were performed using Student's t-test or Mann–Whitney rank sum test.

## Orthologs and paralogs

Proteins encoded by orthologs or close paralogs of CACNA1H in vertebrate and invertebrate species were identified by a BLAST search. GenBank accessions included NP_066921.2 (*Homo sapiens*), O88427.3 (*Mus musculus*), XP_414830.4 (*Gallus gallus*), XP_002932520.2 (*Silurana tropicalis*), XP_002122425.1 (*Ciona intestinalis*) and NP_001024496.1 (*Caenorhabditis elegans*). Human α1 subunit paralogs were as previously described (*Scholl et al., 2013*).

## Principal component analysis, analysis of relatedness, shared haplotypes and mutation age

Principal component analysis was performed as previously described (*Lemaire et al., 2013*). For analysis of close relatedness, genomic DNA from subjects 333-1, 1347-1, 1368-1, 1390-1, and 1393-1 was genotyped on Illumina Human 1M-Quad beadchips according to the manufacturer's instructions. Data were analyzed using a combination of GenomeStudio (Illumina) and PLINK v1.07 softwares (*Purcell et al., 2007*). Mean call rate was 95.7%. Kinship coefficients were calculated by using the robust algorithm in KING 1.4 (*Manichaikul et al., 2010*). For 1393-1, 1368-1 and 333-1, PLINK format was converted to BEAGLE format using Mega2 (*Mukhopadhyay et al., 2005*). Haplotypes flanking the *CACNA1H^{M1549V}* mutation were phased by observed transmission in kindred 1393 and by maximum likelihood in kindreds 1368 and 333 using BEAGLE v.3.3.2 (*Browning and Browning, 2007*) and a reference panel (phase 1 1000Genomes project). Only SNPs called in at least two samples were used for imputation, and only called SNPs were used for determination of the shared interval. Four additional heterozygous variants in close proximity to *CACNA1H^{M1549V}* were identified from the 1393-1 exome. For 1368-1 and 333-1, the inferred haplotype producing the largest shared interval was chosen for further analysis. Mutation age was determined from haplotypes including flanking 41 markers using ESTIAGE (*Genin et al., 2004*). Recombination fractions were calculated from marker distances and average recombination rate across the interval (2.9 cM/Mb, deCODE). Shared allele frequencies were from EUR population (1000 Genomes project), and mutation rate was set to $2 \times 10^{-8}$.

## Statistical analysis

For statistical analysis, a de novo mutation rate of $1.4 \times 10^{-8}$ was assumed. The binomial probability of observing two or more de novo mutations at a specified position in a set of 41 cases (including one affected parent) was calculated and corrected for the target size of the human exome (24.75 Mb). The likelihood of observing three additional independent mutations at the identical position in 38 patients was calculated as a binomial probability from the assumed allele frequency.

The mutation burden per gene in the cohort of patients with PA was compared to that in a control cohort comprising 724 unaffected parents of patients with congenital heart disease sequenced to similar depth of coverage on the same exome platform (*Zaidi et al., 2013*) using Fisher's exact test.

## Acknowledgements

We gratefully acknowledge and thank the patients and families whose participation made this research possible. We thank Dr Patricia Hidalgo, Dr Mathieu Lemaire and Samir Zaidi for helpful discussions, and the staff of the Yale Center for Genome Analysis and Mina Schwenk for technical assistance.

## Additional information

### Funding

| Funder | Grant reference | Author |
| --- | --- | --- |
| National Institutes of Health (NIH) | Centers for Mendelian Genomics, 5U54HG006504 | Richard P Lifton |
| Howard Hughes Medical Institute (HHMI) | Investigator Program | Richard P Lifton |
| Agency for Science, Technology and Research (A*STAR) | Scholarship | Gerald Goh |
| Ministry of Innovation, Science, Research and Technology of the state of North Rhine-Westphalia | Rückkehrerprogramm | Ute I Scholl |

The funders had no role in study design, data collection and interpretation, or the decision to submit the work for publication.

### Author contributions

UIS, Conception and design, Acquisition of data, Analysis and interpretation of data, Drafting or revising the article; GS, MLP, Acquisition of data, Analysis and interpretation of data, Drafting or revising the article; CN-W, AT, Acquisition of data, Analysis and interpretation of data; AAV, ML, Acquisition of data, Drafting or revising the article; MC, CF, Analysis and interpretation of data, Drafting or revising the article; EL, TC, IQ, LCR, BGF, MR, DLB, CBS, HT, Ascertained and recruited subjects, Obtained samples and medical records, Acquisition of data; GG, Analyzed exome sequencing results, Analysis and interpretation of data; CCJ, Ascertained and recruited subjects, Obtained samples and medical records, Prepared DNA samples and maintained sample archives, Acquisition of data; RPL, Conception and design, Analysis and interpretation of data, Drafting or revising the article

### Author ORCIDs

Gabriel Stölting, http://orcid.org/0000-0002-2339-0545

### Ethics

Human subjects: Informed consent for participation was obtained from all research participants, and research protocols were approved by the institutional review board (IRB) at Yale University (8556 protocol, HIC# 9515008556). Specific consent for publication was not obtained because the work does not include identifying, or potentially identifying, information.

## Additional files

### Supplementary file

• Supplementary file 1. (A) Clinical features of 40 patients with primary aldosteronism. (B) Sequencing statistics of 40 exomes. (C) Previously unreported protein-altering variants that occur in more than one subject. (D) Genes with highest burden of rare (<0.01%) heterozygous variants in cases compared to controls. (E) Genes with highest burden of rare (<0.1%) homozygous, hemizygous or candidate compound heterozygous damaging or conserved variants in cases compared to controls. (F) Illumina sequence reads identifying CACNA1H p.Met1549Val in five unrelated subjects. (G) Demonstration of biological parentage by genotyping of short tandem repeat markers in parent-offspring trios in kindreds 1347 and 1390 confirms that *CACNA1H* mutations are de novo in these kindreds. (H) Kinship coefficients of affected individuals from kindreds with *CACNA1H*$^{M1549V}$ variant. (I) Clinical features of family members of index cases with *CACNA1H*$^{M1549V}$.

## Major datasets

The following dataset was generated:

| Author(s) | Year | Dataset title | Dataset ID and/or URL | Database, license, and accessibility information |
|---|---|---|---|---|
| Scholl et al., | 2015 | dbSNP/ClinVar submission | http://www.ncbi.nlm.nih.gov/clinvar/?term=SCV000218508 | Novel disease-associated variant has been deposited in dbSNP (accession number SCV000218508); the patient consent provided does not permit deposition of complete exome data. |

Standard used to collect data: ClinVar submission guidelines have been followed: http://www.ncbi.nlm.nih.gov/clinvar/docs/submit/#where_to_submit.

The following previously published dataset was used:

| Author(s) | Year | Dataset title | Dataset ID and/or URL | Database, license, and accessibility information |
|---|---|---|---|---|
| Exome Aggregation Consortium | 2014 | Data from: ExAC | http://exac.broadinstitute.org | Data are released under a Fort Lauderdale Agreement (http://exac.broadinstitute.org/terms). |

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

## Appendix 1

## Case reports

**K1347**. Subject 1347-1 is a 9-year old male of European ancestry who was diagnosed with hypertension at age 3 years (blood pressures [BPs] 140–160/90–105 mmHg, >99th percentile). Evaluation revealed serum aldosterone 12–20 ng/dl, despite completely suppressed plasma renin activity (PRA) on repeated measures (<0.1 ng/ml/hr); aldosterone:renin ratio (ARR) > 120 to >200 ng/dl:ng/ml/hr. MRI of the abdomen revealed no adrenal or other mass. Further extensive evaluation for other causes of hypertension was unrevealing. He had normal echocardiogram, unremarkable serum chemistry including $K^+$ of 3.8 mmol/l (nl 3.3–4.6), normal renal ultrasound, MR angiogram and computed tomography (CT) angiogram of the renal vessels, normal urine analysis, thyroid function, 24 hr urinary catecholamines, vanillylmandelic acid, homovanillic acid, metanephrines, unremarkable evaluation for congenital adrenal hyperplasia and normal urinary free 18-hydroxy cortisol. The subject was developmentally delayed (walked at age 14 months, spoke at age >2 years) and received physical and occupational therapy in early childhood. Neither parent was hypertensive, and there was no family history of early hypertension, cerebral hemorrhage, stroke or intracranial aneurysm. PRA remained low on treatment with spironolactone and isradipine, and BP was difficult to control. On atenolol, amlodipine and amiloride, he continued to have suppressed PRA, and aldosterone was normal to elevated. His blood pressure is currently well controlled on triple antihypertensive therapy with eplerenone, amlodipine, and chlorthalidone.

**K1390**. Subject 1390-1 is a 15-year old African American female. She was born prematurely at 28 weeks gestational age following a pregnancy complicated by maternal pre-eclampsia and prolonged rupture of membranes (birth weight 1331 g, length 38 cm, head circumference 28 cm, appropriate for gestational age). She developed respiratory distress, bradycardia, mild pulmonary artery branch stenosis, hyperbilirubinemia and necrotizing enterocolitis that responded to medical management. Developmental milestones were reported to be normal. At age 6 years, she was diagnosed with asthma and seasonal allergies. At age 7 years, she presented with headache, blurry vision and hyperactivity, and was diagnosed with hypertension (BP 150/90 mmHg, >99th percentile). Evaluation was remarkable for primary aldosteronism with elevated serum aldosterone (66 ng/dl) and suppressed PRA of 0.2 ng/ml/hr (ARR 330 ng/dl:ng/ml/hr). No adrenal masses or hypertrophy were observed on CT imaging. Renal ultrasound and echocardiography were normal. Follow up revealed hypokalemia (serum $K^+$ 3.1 mmol/l), serum aldosterone of 34 ng/dl and PRA of 0.3 ng/ml/hr (ARR of 113 ng/dl:ng/ml/hr). Her current medications include hydrochlorothiazide, lisinopril, montelukast and loratadine. Her current height is 160 cm (38th percentile), and weight is 68.1 kg (89th percentile). Recent BP off medication was 165/105 mmHg.

The patient's family history was remarkable for early severe hypertension in her 38-year old mother (1390-2) who was diagnosed with hypertension (BP 215/115 mmHg) and hypokalemia at age 17 years. Two pregnancies were complicated by pre-eclampsia, a third resulted in a miscarriage following complication from appendicitis. Evaluation for secondary hypertension was consistent with primary aldosteronism. A CT scan at age 29 years showed mild left adrenal fullness without discrete adrenal mass. Adrenal venous sampling demonstrated aldosterone: cortisol ratios of 92 ng/dl:13.8 µg/dl = 6.67 ng/µg on the left side, 11 ng/dl:6.9 µg/dl = 1.59 ng/µg on the right side and 22 ng/dl:9.7 µg/dl = 2.27 ng/µg in the inferior vena cava. Direct renin was 3 µIU/ml (<5 suggestive of sodium/volume mediated hypertension).

This subject (1390-2) underwent unilateral adrenalectomy at age 29 years to attempt to mitigate severe hypertension with PA. Pathology demonstrated a 4.8 × 3.5 × 2 cm (normal 5 cm × 2.5 cm) specimen weighing 8 g (normal 2.8–5.5 g) (***Neville and O'Hare, 1982***). There was no macroscopic nodularity. Histology was significant for marked hyperplasia of the adrenal zona glomerulosa within a ~0.1 cm cortex with all cell layers and cell types present. While the normal

adrenal glomerulosa comprises only a few cell layers and is about 70 µm in depth, the glomerulosa of subject 1390-2 had ~30 cell layers and was ~300 µm in depth (**Figure 4A–C**). Clusters and trabeculae of glomerulosa cells infiltrated the fibrous capsule of the gland at multiple foci, with occasional micronodules just over the capsule. Staining with specific antibodies demonstrated that *CACNA1H* was expressed in the hypercellular glomerulosa of the patient (**Figure 4D–F**). Consistent with her germline mutation, hypertension recurred after surgery, requiring reinstitution of treatment. Her current medications are hydrochlorothiazide, lisinopril and metoprolol succinate, and recent BP on medication was 138/87.

The index case's father (1390-3), brother (1390-4) and maternal grandfather (1390-10) are normotensive. While the maternal grandmother (1390-5) was diagnosed with adult-onset hypertension, her aldosterone was normal (16 ng/dl), with non-suppressed PRA (18.92 ng/ml/hr) (**Table 2**).

**K1368**. Subject 1368-1 is a 19-year old male of European ancestry. His birth weight was 3740 g. Perinatal history was unremarkable. He underwent pyloromyotomy due to pyloric stenosis at age 5 weeks, and was hospitalized for respiratory distress at age 7 months. At age 8 years, he was found to be hypertensive, and evaluation at age 9 years revealed BPs of 130–140/80–90 mmHg (>99th percentile systolic; >95th percentile diastolic). Serum aldosterone was 20 ng/dl, PRA <0.2 ng/ml/hr (ARR >100 ng/dl:ng/ml/hr), and urinary aldosterone 20 µg/24 hr (nl 2–20). Renal ultrasound and angiogram at age 9 were unremarkable, and no adrenal enlargement was noted. On treatment with lisinopril and amlodipine, BP was 120–130/60–65 mmHg at age 13 (>95th percentile systolic, >50th percentile diastolic). Serum $K^+$ was 3.6 mmol/l, and total $CO_2$ was 30 mmol/l. Repeat aldosterone on medication was 5 ng/dl, and PRA was 0.1 ng/ml/hr. Neurologic evaluation revealed attention deficit disorder, but no medication was started. Height at age 14 was 157.5 cm (10th percentile), and weight was 41 kg (fifth percentile). BP further improved to 118/74 mmHg at age 14 (>50th percentile); serum $K^+$ was 3.8 mmol/l and total $CO_2$ 25 mmol/l. After diagnosis of a mild mitral valve prolapse, treatment with a beta-blocker was started, and amlodipine was discontinued.

The patient's family history was remarkable for a maternal uncle (1368-4) who was diagnosed with hypertension at age 24 (BP 200/100 by history). He was treated with hydrochlorothiazide. At age 51 years, on treatment with 12.5 mg hydrochlorothiazide, his BP was 120/80 mmHg, with aldosterone 11 ng/dl and PRA 1.2 ng/ml/hr (note that treatment with hydrochlorothiazide increases PRA), with serum $K^+$ 3.8 mmol/l. The patient's mother was normotensive, with a home BP of 116/80 mmHg. There was no evidence of primary aldosteronism (aldosterone 8 ng/dl, PRA 4.73 ng/ml/hr, serum $K^+$ 3.7 mmol/l. The maternal grandfather reportedly had hypokalemia and hypertension and a remote history of seizures, which had resolved. He died from hemorrhagic stroke at age 76 years.

**K333**. Subject 333-2 is a 35-year old male of European ancestry who was diagnosed with hypertension at age 9 years (BP 192/144 mmHg) with a history of headaches, enuresis, polyuria and polydipsia. Retrospectively, a BP of 160/80 prior to surgery for unilateral orchiopexy and left inguinal hernia repair at age 2 years was noted. Serum $K^+$ was 3.7 mmol/l, $Cl^-$ 108 mmol/l (nl 95–105), and other serum electrolytes, creatinine and blood urea nitrogen were normal. Extensive evaluation including ultrasound, CT scan, angiogram and i.v. pyelogram did not reveal any renal or adrenal abnormalities. ECG and echocardiogram were normal. Aldosterone levels were 25 and 40 ng/dl, and renin was suppressed at <0.7 ng/ml/hr. A dexamethasone test (0.5 mg 4 times a day) did not change BP, but aldosterone after 36 hr decreased to 3.2 ng/dl, and PRA was 0.3 ng/ml/hr. The subject had normal 24-hr urine vanillylmandelic acid, free cortisol, 5-OH indoleacetic acid and 17-OH progesterone, serum corticosterone, dehydro-piandrosterone, deoxycorticosterone, progesterone, 17-OH pregnenolone, ACTH and TSH. At age 17 years, height was 178 cm (78th percentile), and weight was 75 kg (62th percentile). Treatment with spironolactone, and later minoxidil, atenolol, hydrochlorothiazide, and lisinopril was initiated. At age 35 years, height was 185 cm, and weight was 109 kg. BP was 124/65

mmHg, and serum K$^+$ was 3.7 mmol/l on treatment with atenolol, hydrochlorothiazide, lisinopril, minoxidil, and potassium chloride.

Family history was significant for early-onset hypertension in the subject's father (333-1), who was diagnosed with hypertension at age 13 years, but had been hospitalized at age 5 years for evaluation of precocious puberty, with recorded BPs of 130–160/90–120 mmHg. He has been on antihypertensive medication since age 19 years. The paternal grandmother was also reportedly diagnosed with hypertension in her youth.

**K1393**. Subject 1393-1 is a 5-year old Hispanic male (height 111.8 cm, 72nd %ile, weight 20.9 kg, 82nd %ile). He presented at age 2 months with a viral illness, was admitted to rule out sepsis and found to be extremely hypertensive (BP 170s/100s mmHg, >99th percentile). Serum K$^+$ was 4.1 mmol/l, Cl$^-$ 110 mmol/l, and other electrolytes and creatinine were normal. On echocardiogram, he had mild left ventricular hypertrophy without coarctation. A renal angiogram was normal. Aldosterone was elevated at 87.2 ng/dl, and PRA was suppressed at <0.6 ng/ml/hr. Laboratory evaluation for other causes of hypertension was unrevealing and included normal thyroid function test, 24-hr urine metanephrines, vanillylmandelic acid, homovanillylmandelic acid and tests for congenital adrenal hyperplasia. ECG at age 4 years showed right ventricular conduction delay and prominent mid precordial ventricular forces, and echocardiogram showed minimal concentric left ventricular hypertrophy without left ventricular outflow obstruction, systolic or diastolic dysfunction.

Treatment with propranolol, captopril, hydralazine and spironolactone was started, and aldosterone increased to 290 ng/dl, with renal vein renin <0.6 ng/ml/hr. Current medications include Enalapril, Spironolactone and Propranolol. There was no known family history of hypertension. The father (1393-3) had serum of K$^+$ 3.9 mmol/l, low plasma renin of 0.81 ng/ml/hr and normal aldosterone of 9.3 ng/dl, with BP of 120/82 mmHg.

