## [Decision Letter]

Thank you for sending your work entitled “Recurrent gain of function mutation in *CACNA1H* causes early-onset hypertension with primary aldosteronism” for consideration at *eLife*. Your article has been favorably evaluated by Stylianos Antonarakis (Senior editor) and three reviewers, one of whom, David Ginsburg, is a member of our Board of Reviewing Editors.

The Reviewing editor and the other reviewers discussed their comments before we reached this decision, and the Reviewing editor has assembled the following comments to help you prepare a revised submission.

The authors report results of analyzing whole exome sequence data for 40 subjects with hypertension with primary aldosteronism, identifying five subjects with the same mutation in *CACNA1H*, two of which are confirmed to occur as de novo mutations. The results implicating *CACNA1H* are unequivocal, introducing a new gene for a clinically definable subtype of severe hypertension. Interest in the work is increased by the suggestion that such “cryptic Mendelian” phentoypes may be more widespread than appreciated, given that there are a lot of disparate reasons why relatively highly penetrant mutations many not create discernible segregation patterns, and also by the apparent asymptomatic phenotype in the two parents.

The identified amino acid substitution in Ca_V_3.2 (Calcium channel voltage dependent T type alpha 1H subunit) is in domain III, S6 is in a conserved methionine, which is in a MFV tripeptide shown to regulate inactivation in a related T channel subtype. The data are very strong and clearly show that the mutation delays inactivation significantly, and slightly shifts the activation curve to more hyperpolarized potentials. This is a gain of function and should induce early depolarization and increased calcium flux in cells in which this channel is expressed. Assuming the antibody the authors use is sufficiently specific, high levels are present in human adrenal glomerulosa.

In addressing the surprising finding of two unaffected parents carrying the same mutation, the authors do not address the possibility of post zygotic somatic mosaicism. Could a lower level of mosaicism in the critical tissue (adrenals) in these parents explain the apparent “incomplete penetrance”? Is there any available data for the mutant allele at a ratio less than 0.5, in these parents? Though it may not be practical to conduct additional, more quantitative analyses in these parents, or to survey additional tissues, this possibility should be added to the Discussion.

Minor comments:

1) Though the detailed data and statistical significance are thoroughly covered in the text, the *CACNAIH*^*M1549V*^ mutation is not included in the supplement to Table 1. It could be useful to add these data to the table for direct comparison with the rest of the mutation burden data, and emphasizing the point of how clearly its significance rises above the other identified variants.

2) Though obviously beyond the scope of this manuscript, it will be interesting to know whether loss-of-function mutations in *CACNAIH* are associated with reduced BP in mice or humans. How prevalent are such mutations in the current human whole exome/genome databases?

3) Readers might appreciate, and the field would benefit, from more discussion of why this gain of function mutation does not result in seizures or arrhythmias since *CACNA1H* has been reported to be at high levels in parts of brain, heart, and liver and many allelic variants are reported in OMIM as related to epilepsy (although the authors mention the problem in a previous GWAS study). The distribution of the protein may be much more restricted than previously supposed or the reported Gbeta gamma inhibition of *CACNA1H* may have relative differential importance depending on GPCR type in different tissues. Second, although the authors discount the use of dihydropyridines for therapy, there are non-dihydropyridine T type channel blockers in clinical trials for diabetes and elevated heart rate, and mebefradil has long been used in T channel physiological studies.

---

## [Author Response]

*In addressing the surprising finding of two unaffected parents carrying the same mutation, the authors do not address the possibility of post zygotic somatic mosaicism. Could a lower level of mosaicism in the critical tissue (adrenals) in these parents explain the apparent* “*incomplete penetrance*”*? Is there any available data for the mutant allele at a ratio less than 0.5, in these parents? Though it may not be practical to conduct additional, more quantitative analyses in these parents, or to survey additional tissues, this possibility should be added to the Discussion*.

There are many potential explanations for incomplete penetrance including age-dependence, environmental or genetic modifiers. Among these, somatic mosaicism resulting in absence of the mutation in the adrenal gland is one potential explanation in these families. This could occur if the mutation arose de novo post fertilization, with the mutation being absent in the adrenal gland in non-penetrant subjects. Alternatively, there could be germline transmission of the mutation but somatic loss of the mutation in the adrenal gland due to non-disjunction, mitotic recombination, or second site mutation in *CACNA1H* in adrenal progenitors. Such mosaicism most often is found recurrently when the presence of a mutation in one or more tissues is strongly selected against via lethality or when there is significant selective advantage for stem cells that have lost the disease-causing mutation. Proteus syndrome, Bloom syndrome and Ichthyosis with confetti are examples of diseases in which various of these principles apply.

Of our two non-penetrant individuals, we cannot exclude a de novo mutation with somatic mosaicism in subject 1393-3 since his parents were not available for study; in contrast, subject 1368-2 has an affected, mutation-positive, sibling, and therefore undoubtedly received a gametic mutation, and therefore would need to have achieved mosaicism by somatic loss. Sanger sequencing of peripheral blood (1393-3) or saliva (1368-2) DNA (shown in Figure 1), provided no evidence of a lower prevalence of the mutant allele compared to wildtype, yielding no suggestion of mosaicism. Similarly, Sanger sequencing of peripheral blood DNA in subject 1368-2 did not provide evidence of lower frequency of the mutant allele (see Figure 7). Nonetheless, these findings do not exclude mosaicism in the adrenal gland. Directly addressing this question would require adrenal biopsy, an invasive procedure that does not have medical/ethical justification in these healthy subjects. In the revised manuscript we comment on somatic mosaicism as a possible explanation of incomplete penetrance (in the Discussion section).

Author response image 1.**DOI:**
http://dx.doi.org/10.7554/eLife.06315.020

Minor comments:

*1) Though the detailed data and statistical significance are thoroughly covered in the text, the* CACNAIH^M1549V^
*mutation is not included in the supplement to*
Table 1*. It could be useful to add these data to the table for direct comparison with the rest of the mutation burden data, and emphasizing the point of how clearly its significance rises above the other identified variants*.

We appreciate this point. We performed three analyses of the genomic data-

a search for recurrence of previously unreported variants in cases; analysis of the burden of very rare heterozygous variants in every gene in cases vs. controls; analysis of the burden of rare homozygous/compound heterozygous variants in every gene in cases vs. controls. While we showed the results of the latter two analyses in the supplement to Table 1, we did not show the result of the first analysis since there was only one recurrent previously unreported variant. In the revised manuscript we have added a table showing the results of this analysis (new [Supplementary-material SD4-data]).

*2) Though obviously beyond the scope of this manuscript, it will be interesting to know whether loss-of-function mutations in* CACNAIH *are associated with reduced BP in mice or humans*. *How prevalent are such mutations in the current human whole exome/genome databases?*

In the ExAC database comprising ∼120,000 alleles, the cumulative frequency of likely loss of function mutations (all nonsense, frameshift and canonical splice site variants) was 0.062%, with no homozygous variants reported. Determining the impact of rare loss of function variants on blood pressure will consequently require very large sample sizes. Similarly, assuming Hardy-Weinberg equilibrium, biallelic inactivation is expected in only ∼1 in 2.6 million people. We comment on this in the Discussion section of the revised manuscript.

It is nonetheless interesting to consider whether loss of function mutations in *CACNA1H* will result in lower blood pressure. Calcium channels *CACNA1H* and *CACNA1D* are both expressed in adrenal glomerulosa and seem likely to respectively mediate small increases in aldosterone production in response to small depolarizing stimuli or larger increases in aldosterone in response to larger depolarizing stimuli. Consequently, one might speculate that loss of function mutations in *CACNA1H* could be compensated by secondary increases in activity of the renin-angiotensin system, resulting in increased *CACNA1D* activity, thereby restoring normal aldosterone secretion at increased levels of PRA. An analogous situation obtains in patients who are heterozygous null for the mineralocorticoid receptor (the nuclear hormone receptor that is normally activated by aldosterone). These patients maintain normal blood pressure via increased activity of the renin-angiotensin system and markedly elevated aldosterone levels, restoring signaling through the mineralocorticoid receptor to near-normal levels. Some support for this possibility comes from the *cacna1h* knockout mouse, which has been reported to have normal blood pressure (12); it would be interesting to evaluate the levels of plasma renin activity and aldosterone in these mice, which to our knowledge have not been published.

*3) Readers might appreciate, and the field would benefit, from more discussion of why this gain of function mutation does not result in seizures or arrhythmias since* CACNA1H *has been reported to be at high levels in parts of brain, heart, and liver and many allelic variants are reported in OMIM as related to epilepsy (although the authors mention the problem in a previous GWAS study). The distribution of the protein may be much more restricted than previously supposed or the reported Gbeta gamma inhibition of* CACNA1H *may have relative differential importance depending on GPCR type in different tissues. Second, although the authors discount the use of dihydropyridines for therapy, there are non-dihydropyridine T type channel blockers in clinical trials for diabetes and elevated heart rate, and mebefradil has long been used in T channel physiological studies*.

We appreciate these considerations. Just as guessing which genes might be mutated from consideration of phenotypes observed in patients is frequently wrong, inferring what phenotypes will result from a given mutation is often incorrect, reflecting our incomplete understanding of human biology. The observed human phenotypes resulting from a specific mutation directly address the question and leave us to explain the results a posteriori. None of our patients with the *CACNA1H*^M1549V^ mutation have a history of seizures or cardiac arrhythmia. This includes two subjects who had formal neurologic evaluation. Subject 1368-1 was evaluated at age 10 for learning disability; subject 1347-1 was evaluated at age 4 for developmental delay (which has subsequently resolved). These evaluations revealed no evidence of seizures in either subject including an EEG in subject 1368-1, which was interpreted as normal. There does not appear to be cognitive or other neurologic impairment in any mutation carriers.

There are many plausible reasons that gain of function mutations in *CACNA1H* might have effects in adrenal but not other tissues. Aldosterone secretion in adrenal glomerulosa appears to be a direct consequence of increased intracellular calcium. As noted in the manuscript, *CACNA1H* produces relatively modest calcium influx in response to small depolarizing potentials. This might be sufficient to increase aldosterone secretion but not achieve the threshold levels or rapid changes in membrane potential required for epilepsy. Alternatively, *CACNA1H* could play a role in inhibitory interneurons, and gain of function could result in increased inhibition. The neurologic phenotypes in patients with these *CACNA1H* mutations stand in contrast to those resulting from recurrent gain of function mutations in another calcium channel, *CACNA1D*. This channel is activated by larger depolarizing potentials and appears to have larger single channel conductance (42; 5). In addition to aldosteronism, patients with germline gain of function mutations in *CACNA1D* also have seizures, severe intellectual impairment and neuromuscular defects (55). As noted by the reviewers, *CACNA1H* appears to be inhibited by G-protein beta-gamma subunits, raising the further possibility that differential use of beta-gamma signaling in different tissues could contribute to differential activity.

Similarly, *cacna1h* knockout mice do not show cardiac arrhythmias or ECG abnormalities, and no T-type currents are expressed in adult ventricular myocytes under physiological conditions (10), arguing against a key role of the channel in pacemaker activity or conduction in the heart.

An association of rare *CACNA1H* variants with absence and other epilepsy phenotypes in humans has been reported (10; 28; 37), however there are caveats to these studies. These results have not approached genome-wide levels of significance, and some studies have not found evidence of increased burden of the initially reported rare mutations in *CACNA1H* in epilepsy phenotypes (27; 13). The effects on channel activity, when measured, have been small (smaller than the effect observed with the mutation found in our study) (Khosravani et al., 2004; Eckle et al., 2014; Khosravani et al., 2005). None of the identified variants are de novo mutations, and they show no significant evidence of linkage with epilepsy phenotypes within kindreds, leading to the suggestion that these might impart small effects on epilepsy risk (Eckle et al., 2014). No association of common variants in or near *CACNA1H* with epilepsy in general or absence epilepsy specifically has been detected by GWAS analysis, including a recent meta-analysis of 8696 cases and 26157 controls and a study of 3020 epilepsy patients and 3954 controls of European origin (Genetic determinants of common epilepsies: a meta-analysis of genome-wide association studies. Lancet Neurol. 2014;13(9):893-903; Genome-wide association analysis of genetic generalized epilepsies implicates susceptibility loci at 1q43, 2p16.1, 2q22.3 and 17q21.32. Hum Mol Genet. 2012;21(24):5359-72). On balance, the role of variants in *CACNA1H* in epilepsy seems uncertain.

Regarding other calcium channel blockers, Mibefradil, which was withdrawn from the market due to unfavorable drug interactions, is a potent inhibitor of *CACNA1H* in vitro (47) and inhibited aldosterone production in bovine adrenal glomerulosa in vitro (52), but no sustained reduction in aldosterone was seen upon systemic administration to healthy volunteers (50) or in hypertensive patients (53). The results of ongoing clinical studies of non-dihydropyridine calcium channel blockers on blood pressure, and aldosterone in particular, remain to be reported. In the revised manuscript, we comment on these studies in the Discussion.